# SYNC: Safety-Aware Neural Control for Stabilizing Stochastic Delay-Differential Equations

**Jingdong Zhang**[1,2]**, Qunxi Zhu**[2,*]**, Wei Yang**[2,3,*]**, Wei Lin**[1,2,3,4*]

[1] School of Mathematical Sciences, SCMS, SCAM, and CCSB, Fudan University,
  Shanghai 200433, China
[2] Research Institute of Intelligent Complex Systems, Fudan University, Shanghai 200433, China
[3] Shanghai Artificial Intelligence Laboratory, China
[4] MOE Frontiers Center for Brain Science and State Key Laboratory of Medical Neurobiology,
  Fudan University, Shanghai 200032, China

{zhangjd20,qxzhu16,yangwei,wlin}@fudan.edu.cn

## Abstract

Stabilization of the systems described by *stochastic delay*-differential equations (SDDEs) under preset conditions is a challenging task in the control community. Here, to achieve this task, we leverage neural networks to learn control policies using the information of the controlled systems in some prescribed regions. Specifically, two learned control policies, i.e., the neural deterministic controller (NDC) and the neural stochastic controller (NSC), work effectively in the learning procedures that rely on, respectively, the well-known LaSalle-type theorem and the newly-established theorem for guaranteeing the stochastic stability in SDDEs. We theoretically investigate the performance of the proposed controllers in terms of convergence time and energy cost. More practically and significantly, we improve our learned control policies through considering the situation where the controlled trajectories only evolve in some specific safety set. The practical validity of such control policies restricted in safety set is attributed to the theory that we further develop for safety and stability guarantees in SDDEs using the stochastic control barrier function and the spatial discretization. We call this control as SYNC (**S**afet**Y**-aware **N**eural **C**ontrol). The efficacy of all the articulated control policies, including the SYNC, is demonstrated systematically by using representative control problems.

## 1 Introduction

Stochastic delay-differential equations (SDDEs) (Mao, 1996; Lin & He, 2005; Sun & Cao, 2007; Guo et al., 2016) have been widely applied to characterize the complex dynamical behavior emergent in real-world systems with dependence on the current state, the past state, and the noise. Efficiently controlling these systems is a long-standing and crucial problem, with the consequent emphasis being placed on the design of control policies and analysis of stability in SDDEs. Traditional control methods in stochastic settings have been fully developed in the convex optimization frameworks using the control Lyapunov stability theory, e.g. the quadratic programming (QP) (Fan et al., 2020; Sarkar et al., 2020). These methods cannot provide the analytical form of feedback controllers and own a high computational cost, requiring solving QP problems at each iteration step. To overcome these difficulties, utilizing neural networks (NNs) to automatically design controllers becomes one of the mainstream approaches in recent years (Zhang et al., 2022; Chang et al., 2019). However, existing machine-learning-based methods either focus on controlling systems without time-delay or aim at learning the control Lyapunov function instead of the control policy (Khansari-Zadeh &

---
[*]To whom correspondence should be addressed: Q.Z., W.Y. and. W.L.,
https://faculty.fudan.edu.cn/wlin/zh_CN/index.htm.

Billard, 2014). All these, therefore, motivate us to design neural controllers for general nonlinear SDDEs.

The safety verification of controlled systems plays an important role in many branches of cybernetics and industry. For example, with the safety verification, one can reduce a significant economic burden and loss of life (Ames et al., 2016; Wang et al., 2016). In particular, the dominant framework for safety control in stochastic settings is the use of stochastic control barrier function (SCBF) (Clark, 2019; 2021; Santoyo et al., 2021). The core idea of designing a candidate SCBF is that its value tends to explode as the system's state leaves the safe region, implying a safety guarantee as long as one could design a controller such that the SCBF is always finite within the controlled time duration. Unfortunately, the existing theories of SCBF either require a lot of inequality constraints or are limited in handling systems without any time delay.

In this paper, we utilize neural networks (NNs) to learn control policies for SDDEs based on the corresponding stability theories. Additionally, we develop a simplified SCBF theory for SDDEs and then use it to construct the neural controller with a safety guarantee, named SYNC. All these control policies are intuitively depicted in Figure 1. The major contributions of this paper include:

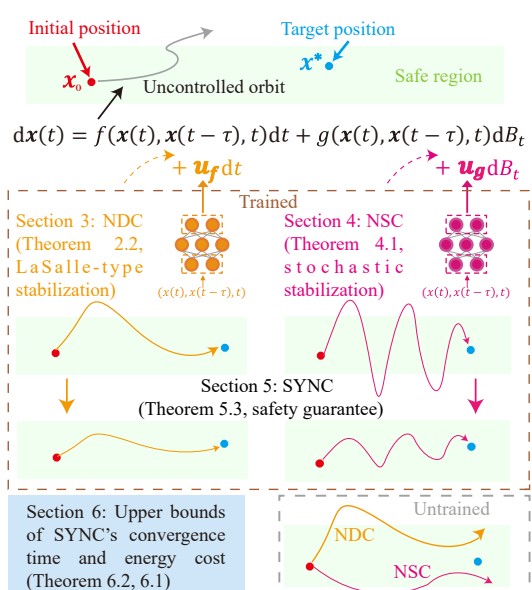

Figure 1: Overall work flow. Sketches of SYNC. Both the NDC and NSC can stabilize the SDDEs to the target unstable equilibrium $\boldsymbol{x}^*$. The safety-aware controlled state trajectories are restricted in the safe region.

- designing a novel and practical framework of neural deterministic control based on the existing LaSalle-Type stability theory,

- proposing a simplified stability theorem and designing the second novel neural stochastic control framework that can benefit from noise according to this theorem,

- establishing an SCBF theory for SDDEs as well as a theory of safety guarantee and stability guarantee using neural network settings,

- providing theoretical estimation for the proposed neural controller in terms of convergence time and energy cost based on the developed theory of safety and stability guarantees, and

- demonstrating the efficacy of the proposed neural control methods through numerical comparisons with the typical existing control methods on several representative physical systems.

## 2 PRELIMINARIES

To begin with, we consider the SDDE in a general form of

$$\mathrm{d}\boldsymbol{x}(t) = F(\boldsymbol{x}(t), \boldsymbol{x}(t-\tau), t)\mathrm{d}t + G(\boldsymbol{x}(t), \boldsymbol{x}(t-\tau), t)\mathrm{d}B_t, \ t \geq 0, \ \tau > 0, \ \boldsymbol{x}(t) \in \mathbb{R}^d, \quad (1)$$

where $\boldsymbol{x}(t) = \xi(t) \in C_{\mathcal{F}_0}([-\tau, 0]; \mathbb{R}^d)$ is the initial function, the drift term $F : \mathbb{R}^d \times \mathbb{R}^d \times \mathbb{R}_+ \to \mathbb{R}^d$ and the diffusion term $G : \mathbb{R}^d \times \mathbb{R}^d \times \mathbb{R}_+ \to \mathbb{R}^{d \times r}$ are Borel-measurable functions, and $B_t$ is a standard $r$-dimensional ($r$-D) Brownian motion defined on probability space $(\Omega, \mathcal{F}, \{\mathcal{F}_t\}_{t \geq 0}, \mathbb{P})$ with a filtration $\{\mathcal{F}_t\}_{t \geq 0}$ satisfying the regular conditions. Without loss of generality, we assume that $F(\mathbf{0}, \mathbf{0}, t) = \mathbf{0}$ and $G(\mathbf{0}, \mathbf{0}, t) = \mathbf{0}$. This assumption guarantees that the zero solution $\boldsymbol{x}(t) \equiv \mathbf{0}$ with $t \geq 0$ is an equilibrium of Eq. (1). Additionally, the following notations and assumptions are used throughout the paper.

**Assumption 2.1** *Assume that Eq. (1) has a unique solution $\boldsymbol{x}(t, \xi)$ on $t \geq 0$ for any $\xi \in C_{\mathcal{F}_0}([-\tau, 0]; \mathbb{R}^d)$ and that, for every integer $n \geq 1$, there is a number $K_n > 0$ such that*

$$\|F(\boldsymbol{x}, \boldsymbol{y}, t)\| \vee \|G(\boldsymbol{x}, \boldsymbol{y}, t)\|_{\mathrm{F}} \leq K_n,$$

*for any $(\boldsymbol{x}, \boldsymbol{y}, t) \in \mathbb{R}^d \times \mathbb{R}^d \times \mathbb{R}_+$ with $\|\boldsymbol{x}\| \vee \|\boldsymbol{y}\| \leq n$, where $\|\cdot\|$ denotes the $L^2$-norm and $\|\cdot\|_{\mathrm{F}}$ denotes the Frobenius norm, i.e. $\|G(\boldsymbol{x}, \boldsymbol{y}, t)\|_{\mathrm{F}}^2 = \sum_{i=1}^d \sum_{j=1}^r G_{ij}(\boldsymbol{x}, \boldsymbol{y}, t)^2$.*

**Definition 2.1** *(Derivative Operator) Define the differential operator $\mathcal{L}$ associated with Eq. (1) by*

$$\mathcal{L} \triangleq \frac{\partial}{\partial t} + \sum_{i=1}^d F_i(\boldsymbol{x}, \boldsymbol{y}, t)\frac{\partial}{\partial x_i} + \frac{1}{2}\sum_{i,j=1}^d [G(\boldsymbol{x}, \boldsymbol{y}, t)G^\top(\boldsymbol{x}, \boldsymbol{y}, t)]_{ij}\frac{\partial^2}{\partial x_i \partial x_j}.$$

According to the above definition of the derivative operator, an operation of $\mathcal{L}$ on the function $V \in C^{2,1}(\mathbb{R}^d \times \mathbb{R}_+; \mathbb{R})$ yields:

$$\mathcal{L}V(\boldsymbol{x}, \boldsymbol{y}, t) = V_t(\boldsymbol{x}, t) + \nabla V(\boldsymbol{x}, t)^\top F(\boldsymbol{x}, \boldsymbol{y}, t) + \frac{1}{2}\mathrm{Tr}\left[G^\top(\boldsymbol{x}, \boldsymbol{y}, t)\mathcal{H}V(\boldsymbol{x}, t)G(\boldsymbol{x}, \boldsymbol{y}, t)\right]. \quad (2)$$

Here, $V_t$, $\nabla V$ and $\mathcal{H}V$ represent, respectively, the time derivative, the gradient, and the Hessian matrix of $V$. Notably, the following LaSalle-type stability theorem will be crucial to the establishment of our partial results.

**Theorem 2.2** *(Mao, 2002) Suppose that Assumptions 2.1 holds. Assumes there are functions $V \in C^{2,1}(\mathcal{X} \times \mathbb{R}_+; \mathbb{R}_+)$, $\gamma \in L^1(\mathbb{R}_+; \mathbb{R}_+)$, and $w_1, w_2 \in C(\mathcal{X}; \mathbb{R}_+)$ such that $\mathcal{L}V(\boldsymbol{x}, \boldsymbol{y}, t) \leq \gamma(t) - w_1(\boldsymbol{x}) + w_2(\boldsymbol{y})$, $w_1(\boldsymbol{x}) \geq w_2(\boldsymbol{x})$, and $\lim_{\|\boldsymbol{x}\| \to \infty} \inf_{0 \leq t \leq \infty} V(\boldsymbol{x}, t) = \infty$. Here, $\mathcal{X} \subset \mathbb{R}^d$ is the state space. Then, $\mathrm{Ker}(w_1 - w_2) \neq \varnothing$ and $\lim_{t \to \infty} \mathrm{dist}(\boldsymbol{x}(t, \xi), \mathrm{Ker}(w_1 - w_2)) = 0$ a.s., where $\mathrm{Ker}(w_1 - w_2) \triangleq \{\boldsymbol{x} : w_1(\boldsymbol{x}) - w_2(\boldsymbol{x}) = 0\}$, $\mathrm{dist}(x, K) \triangleq \inf_{y \in K} \|x - y\|$ for a set $K \subseteq \mathbb{R}^d$, and a.s. stands for the abbreviation of almost surely.*

**Problem Statement**  We assume that the zero solution of the following SDDE:

$$\mathrm{d}\boldsymbol{x}(t) = f(\boldsymbol{x}, \boldsymbol{x}(t - \tau), t)\mathrm{d}t + g(\boldsymbol{x}, \boldsymbol{x}(t - \tau), t)\mathrm{d}B_t \quad (3)$$

is unstable, i.e. $\lim_{t \to \infty} \boldsymbol{x}(t; \xi) \neq \boldsymbol{0}$ on some set of positive measures. We aim to stabilize the zero solution using control based on neural networks (NNs). In other words, our goal is to leverage the NNs to design an appropriate controller $\boldsymbol{u} = (\boldsymbol{u}_f, \boldsymbol{u}_g)$ with $\boldsymbol{u}_f(\boldsymbol{0}, \boldsymbol{0}, t) = \boldsymbol{u}_g(\boldsymbol{0}, \boldsymbol{0}, t) = \boldsymbol{0}$ such that the controlled system

$$\mathrm{d}\boldsymbol{x} = [f + \boldsymbol{u}_f(\boldsymbol{x}(t), \boldsymbol{x}(t - \tau), t)]\mathrm{d}t + [g + \boldsymbol{u}_g(\boldsymbol{x}(t), \boldsymbol{x}(t - \tau), t)]\mathrm{d}B_t \quad (4)$$

is steered to the zero solution. We call $\boldsymbol{u}_f : \mathbb{R}^d \times \mathbb{R}^d \times \mathbb{R}_+ \to \mathbb{R}^d$ as deterministic control while we call $\boldsymbol{u}_g : \mathbb{R}^d \times \mathbb{R}^d \times \mathbb{R}_+ \to \mathbb{R}^{d \times r}$ as stochastic control, since they are integrated with $\mathrm{d}t$ and $\mathrm{d}B_t$, respectively. The major difficulty of this problem comes from the non-Markovian property of SDDEs. As such, we cannot apply the Markov decision process (MDP)-based methods, such as the reinforcement learning, to control SDDEs. The majority of existing works prefer to learn deterministic control and often regard the noise as a negative ingredient that may destroy the natural dynamics of $f$. In what follows, we not only show that the deterministic control can achieve stabilization in a probability sense, but also that elaborately-designed stochastic control can make the same stabilization. This, therefore, yields two frameworks, viz., the neural deterministic control (Section 3) and the neural stochastic control (Section 4). We make all our code and data available at `https://github.com/jingddong-zhang/SYNC`.

## 3    NEURAL DETERMINISTIC CONTROL

In this section, we propose the neural deterministic controller (NDC) based on the Theorem 2.2 to stabilize system (3). Heuristically, we construct the neural network form auxiliary functions and control functions, and integrate the sufficient conditions in the theorem into the loss function to find the neural controller that satisfies the expected conditions. However, the NDC can neither be used to find stochastic controllers nor rigorously satisfy the expected stability conditions. These problems will be addressed in Section 4 and 5.

## 3.1 METHOD: LEARNING CONTROL AND AUXILIARY FUNCTIONS

The core idea of our method is base on using Theorem 2.2, that is, once we construct the auxiliary functions $V$, $\gamma$, $w_1$, $w_2$ and the neural controller $\boldsymbol{u}$ to meet all the conditions assumed in Theorem 2.2 for the controlled system (4), the solution $\boldsymbol{x}(t;\xi)$ converges to the $\mathrm{Ker}(w_1 - w_2)$. In particular, if we set $\mathrm{Ker}(w_1 - w_2) = \{\boldsymbol{0}\}$, the unstable zero solution of the control-free system (3) can be stabilized. To this end, we first provide appropriate constructions of NNs to learn these candidate functions. Thus, we design the explicit form of the loss function in the learning step.

**Auxiliary Function**   We employ a multi-layer feedforward neural network, denoted by $\mathbf{NN}(\cdot;\theta)$, to design all the functions. Precisely, $\theta_1$ is the parameter vector of the positive function $V(\boldsymbol{x}, t; \theta_1)$, and the $L_2$ term $\|\boldsymbol{x}\|^2$ is added to guarantee $\lim_{\|\boldsymbol{x}\|\to\infty} \inf_{0\le t<\infty} V(\boldsymbol{x}, t; \theta_1) = \infty$, that is

$$V(\boldsymbol{x}, t; \theta_V) = \mathbf{NN}(\boldsymbol{x}, t; \theta_V)^2 + \varepsilon\|\boldsymbol{x}\|^2, \ \ \varepsilon > 0. \tag{5}$$

In our framework, it requires $V \in C^{2,1}(\mathbb{R}^d \times \mathbb{R}_+)$. We therefore use a $C^2$ activation function for an NN, such as the hyperbolic tangent function, $\mathrm{Tanh}(\cdot)$. We further discuss the impact of the $L_2$ term in Appendix A.1.3. In order to design an integrable positive function $\gamma(t)$ with the NN, we use an activation function with at most linear growth such as ReLU and multiply an exponential decay factor to the output of the NN, that is

$$\gamma(t; \theta_\gamma) = \exp(-ct) \cdot \mathbf{NN}(t; \theta_\gamma)^2, \ \ c > 0. \tag{6}$$

For simplicity, we design $w(\boldsymbol{x}, \theta_w) = \mathbf{NN}(\boldsymbol{x}; \theta_w)^2$ as a positive function. Additionally, we set

$$w_2 = w, \ \ w_1 = w + p(x), \ \ p \ge 0, \ \ \mathrm{Ker}(p) = \{\boldsymbol{0}\}. \tag{7}$$

**Deterministic Control Function**   We first consider the deterministic control, i.e. $\boldsymbol{u} = (\boldsymbol{u}_f, \boldsymbol{0})$. To guarantee the same zero solution of the control-free system (3) and the controlled system (4), the NDC $\boldsymbol{u}_f : \mathbb{R}^d \times \mathbb{R}^d \times \mathbb{R}^+ \to \mathbb{R}^d$ should satisfy $\boldsymbol{u}_f(\boldsymbol{0}, \boldsymbol{0}, t) = \boldsymbol{0}$. One feasible way to meet such a condition is to set $\boldsymbol{u}_f(\boldsymbol{x}, \boldsymbol{y}, t) = \mathbf{NN}(\boldsymbol{x}, \boldsymbol{y}, t; \theta_f) - \mathbf{NN}(\boldsymbol{0}, \boldsymbol{0}, t; \theta_f)$ or $\boldsymbol{u}_f(\boldsymbol{x}, \boldsymbol{y}, t) = \mathrm{diag}(\boldsymbol{x})\mathbf{NN}(\boldsymbol{x}, \boldsymbol{y}, t; \theta_f)$. Here, $\mathrm{diag}(\boldsymbol{x})$ is a diagonal matrix with $x_i$ as its $i$-th diagonal element.

**Loss Function**   Once the learned functions $V, \gamma, w_1, w_2$ and $\boldsymbol{u}$ with the coefficient functions, $f_{\boldsymbol{u}} \triangleq f + \boldsymbol{u}$ and $g$, in the controlled system (4), meet all the conditions assumed in Theorem 2.2, the stability of zero solution is naturally assured. To achieve this, we demand a suitable loss function to evaluate the likelihood that those conditions are satisfied. It can be seen from our construction that the only condition needed to be satisfied is $\mathcal{L}V(\boldsymbol{x}, \boldsymbol{y}, t) \le \gamma(t) - w_1(x) + w_2(y)$. Hence, we define LaSalle's loss function for the controlled system (4) as follows.

**Definition 3.1** *(LaSalle's Loss)* Consider the above parameterized candidate functions $V, \gamma, w_1, w_2$ and a controller $\boldsymbol{u}_f$ for the controlled system (4). Then, LaSalle's loss is defined as

$$L_{N,\varepsilon,c,p}(\boldsymbol{\theta}_V, \boldsymbol{\theta}_\gamma, \boldsymbol{\theta}_w, \boldsymbol{\theta}_f) \ = \ \frac{1}{N}\sum_{i=1}^{N} \max\left(0, \mathcal{L}V(\boldsymbol{x}_i, \boldsymbol{y}_i, t_i) - \gamma(t_i) + w_1(\boldsymbol{x}_i) - w_2(\boldsymbol{y}_i)\right), \tag{8}$$

where $\{\boldsymbol{x}_i, \boldsymbol{y}_i, t_i\}_{i=1}^{N}$ are sampled from some distribution $\mu$ on $\mathbb{R}^d \times \mathbb{R}^d \times \mathbb{R}_+$.

In summary, the developed NDC framework is shown in Algorithm 1 in Appendix A.3.1.

**Remark 3.1** *The proposed NDC framework can be easily applied to the autonomous SDDE:* $\mathrm{d}\boldsymbol{x}(t) = f(\boldsymbol{x}, \boldsymbol{x}(t - \tau))\mathrm{d}t + g(\boldsymbol{x}, \boldsymbol{x}(t - \tau))\mathrm{d}B_t$. *In particular, one can simply consider the autonomous auxiliary function $V$ and the control function, and set $\gamma(t) = 0$. For sample distribution $\mu(\Omega)$, here we select the uniform distribution on a sufficiently large and closed region $\Omega$ as used in (Han et al., 2016; Chang et al., 2019), and we include further analyses for the impact of $\mu$ in Appendix A.2.1.*

## 3.2 NUMERICAL AND ANALYTICAL INVESTIGATIONS

**Comparison Studies**   Recent works on controlling time-delayed systems mainly focus on elaborately designing the analytical form of control to satisfy the conditions in the LaSalle-Type Theorem 2.2 (Lin & He, 2005; Xu et al., 2014), or simultaneously designing control and the Lyapunov function

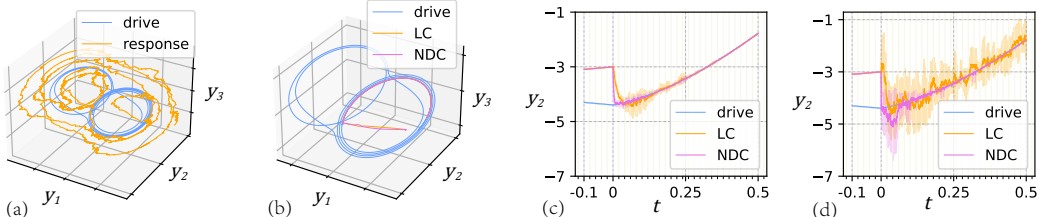

Figure 3: (a) The original driving-response model, (b) the controlled orbits under LC and NDC, (c) the time trajectory of $y_2$ with autonomous noise, and (d) the nonautonomous noise. The solid lines are obtained through averaging the 10 sampled trajectories, while the shaded areas stand for the standard errors.

to satisfy the conditions based on the Lyapunov theory (Yu & Cao, 2007). It should be noted that all these methods require a delicate design of functions for specific dynamics, and thus are limited in practical application for controlling general time-delayed systems. However, our neural method leverages NNs to automatically learn the control policies, and can be applied in any kind of time-delayed systems with stochastic settings. In Figure 3, we numerically compare the NDC and a baseline, the linear control (LC) proposed in (Lin & He, 2005), on a noised driving-response Chua's circuit. Here, Chua's circuit is a three-dimensional autonomous dynamical system with a unique nonlinear element, producing typical chaotic dynamics (Matsumoto, 1984). In the simulation, we show that the NDC can find the neural control for the response system $\boldsymbol{y} = (y_1, y_2, y_3)$ with the autonomous and even the nonautonomous time-delay noise. Actually, the nonautonomous time-delay noise was not considered in (Lin & He, 2005). The simulation configurations are described in Appendix A.3.4.

**Failure in Finding Stochastic Control**   As we can see that the NDC performs well, a natural idea is to utilize the noise part to achieve the stabilization of the SDDE (3). To explore this idea, we adopt the same NN of $\boldsymbol{u}_f$, design $\boldsymbol{u}_g = \mathbf{NN}(\boldsymbol{x}, \boldsymbol{y}, t; \boldsymbol{\theta}_g)$, and train its parameters $\boldsymbol{\theta}_g$ with LaSalle's

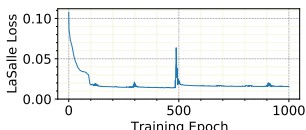

Figure 2: Training loss for 1-D SDDE.

loss (8). However, in Figure 2, we show that the loss cannot converge to zero in controlling a simple 1-D toy system via the stochastic controller $\boldsymbol{u}_g$: $\mathrm{d}x(t) = [x(t) + x(t-\tau)]\mathrm{d}t + [x(t-\tau) + u_g(x(t), x(t-\tau); \theta_g)]\mathrm{d}B_t$. Actually, this phenomenon can be analytically explained. Notice that $\boldsymbol{\theta}_g$ arises in loss function as a quadratic term $l(\boldsymbol{\theta}_g) = \frac{1}{2}\mathrm{Tr}[\boldsymbol{u}_g^\top \mathcal{H}V \boldsymbol{u}_g]$ according to Eq. (2), the sign of this term depends on the convexity of $V$, i.e. the maximum eigenvalue's sign of $\mathcal{H}V$. Nevertheless, the positive function $V$ with $\lim_{\|\boldsymbol{x}\| \to \infty} V(\boldsymbol{x}, t) = \infty$ implies $l(\boldsymbol{\theta}_g) \geq 0$ for most of time. Hence, when we minimize $l(\boldsymbol{\theta}_g) \geq 0$ in the training procedure, the ideal case $l(\boldsymbol{\theta}_g) = 0$ is equivalent to $\boldsymbol{u}_g = 0$. This indicates that we are unable to learn a stochastic controller under LaSalle's loss (8) satisfying the sufficient conditions assumed in Theorem 2.2.

## 4   NEURAL STOCHASTIC CONTROL

To find the neural stochastic controller (NSC), we provide the following theoretical result on stabilization of general stochastic functional differential equations (SFDEs) with the proof provided in Appendix A.1.4. Since the failure of NDC in the stochastic control case comes from the positive number contributed by the diffusion term, we aim at constructing stability condition such that the part related to the diffusion term can be negative. We further explain the Theorem 4.1 in Appendix A.1.4.

**Theorem 4.1** (Stochastic Stabilization) *Consider the SFDE* $\mathrm{d}\boldsymbol{x}(t) = F(\boldsymbol{x}_t, t)\mathrm{d}t + G(\boldsymbol{x}_t, t)\mathrm{d}B(t)$, *with* $F, G$ *being locally Lipschitzian functions,* $F(\boldsymbol{0}, t) = \boldsymbol{0}$, *and* $G(\boldsymbol{0}, t) = \boldsymbol{0}$. *For every* $M > 0$, *assume that* $\min_{\|\boldsymbol{x}_t(0)\|=M} \|\boldsymbol{x}_t(0)^\top G(\boldsymbol{x}_t, t)\| > 0$. *If there exists a number* $\alpha \in (0, 1)$ *such that*

$$\|\boldsymbol{x}_t(0)\|^2 (2\langle \boldsymbol{x}_t(0), F(\boldsymbol{x}_t, t)\rangle + \|G(\boldsymbol{x}_t, t)\|_\mathrm{F}^2) - (2 - \alpha)\|\boldsymbol{x}_t(0)^\top G(\boldsymbol{x}_t, t)\|^2 \leq 0, \qquad (9)$$

*for* $\boldsymbol{x}_t \in C([-\tau, 0], \mathcal{X})$, *where* $\boldsymbol{x}_t(s) = \boldsymbol{x}(t + s)$ *for* $s \in [-\tau, 0]$ *and* $\mathcal{X}$ *is the state space. Then, the solution of the SFDE satisfies* $\lim_{t \to \infty} \boldsymbol{x}(t; \xi) = \boldsymbol{0}$ *a.s. for any* $\xi \in C_{\mathcal{F}_0}([-\tau, 0]; \mathbb{R}^d)$.

**Remark 4.2** *The SFDE in Theorem 4.1 is formulated in a very general type, including the SDDE* $\mathrm{d}\boldsymbol{x}(t) = F(\boldsymbol{x}(t), \boldsymbol{x}(t - \tau_1), \cdots, \boldsymbol{x}(t - \tau_q), t)\mathrm{d}t + G(\boldsymbol{x}(t), \boldsymbol{x}(t - \tau_1), \cdots, \boldsymbol{x}(t - \tau_q), t)\mathrm{d}B_t$ *with*

$\tau_1 < \tau_2 < \cdots < \tau_q \in [0, \tau]$. *This indicates that our framework can be generalized to stabilize the SDDEs with multiple delays and even more general SFDEs as well.*

In light of Theorem 4.1, we establish a more general framework for learning a neural controller of system (4) with the form $\boldsymbol{u} = (\boldsymbol{u}_f, \boldsymbol{u}_g)$ designed in the same NN architecture as the one used in the NDC framework. We focus on stochastic control with $\boldsymbol{u}_f = \boldsymbol{0}$ and provide more control combinations in Appendix A.3.3, whereas the loss function is differently designed as follows.

**Definition 4.1** *(Asymptotic Loss)* Utilize the notations set in Definition 3.1 and $g_{\boldsymbol{u}} = g + \boldsymbol{u}_g$. The loss function for the controlled system (4) with the controller $\boldsymbol{u}$ is defined as:

$$L_{\mu,\alpha}(\boldsymbol{\theta}) = \frac{1}{N} \sum_{i=1}^{N} \left[ \max \left( 0, (\alpha - 2) \|\boldsymbol{x}_i^\top g_{\boldsymbol{u}}(\boldsymbol{x}_i, \boldsymbol{y}_i, t_i)\|^2 + \|\boldsymbol{x}_i\|^2 (2\langle \boldsymbol{x}_i, f(\boldsymbol{x}_i, \boldsymbol{y}_i, t_i)\rangle + \|g_{\boldsymbol{u}}(\boldsymbol{x}_i, \boldsymbol{y}_i, t_i)\|_{\mathrm{F}}^2)) \right], \tag{10}$$

where $\boldsymbol{\theta} = (\boldsymbol{\theta}_f, \boldsymbol{\theta}_g)$. Akin to Definition 3.1, we use the empirical loss function for training.

Here, $\alpha$ is an adjustable parameter, which is related to the convergence rate and the control energy. We further discuss the design of the asymptotic loss in Appendix A.2.2 and numerically investigate the role of $\alpha$ in Appendix A.4.1. We summarize the framework in Algorithm 2 in Appendix A.3.1. And we further compare the computational complexity in Appendix A.3.2.

## 4.1 EXPERIMENTS OF THE COMBINATION METHODS

We compare our neural control methods on a noise-perturbed kinematic bicycle model for car-like vehicles (Rajamani, 2011) in terms of the convergence time and the energy cost, which are two important indexes to measure the quality of a controller (Yan et al., 2012; Li et al., 2017; Sun et al., 2017). To

Table 1: Results on kinematic bicycle model.

|     | Tt | $\mathcal{E}_{0.001}$ | Nd | $\mathbb{E}[\tau_{0.001}]$ |
|-----|-----|------------|------|--------------|
| NDC | 1028.81s | 102.17 | 6.3e-4 | 1.81 |
| NSC | **59.80**s | **62.10** | 4.0e-7 | 0.29 |
| QP | - | - | 0.016 | > 5 |

quantify the energy cost in the control process, we first denote by $\tau_\epsilon \triangleq \inf\{t > 0 : \|\boldsymbol{x}(t)\| = \epsilon\}$ the stopping time and then by $\mathcal{E}_\epsilon \triangleq \mathbb{E}\left[\int_0^{\tau_\epsilon} (\|\boldsymbol{u}_f\|^2 + \|\boldsymbol{u}_g\|^2)\,\mathrm{d}t\right]$ the energy cost. We approximate this expectation value by the empirical value as $\frac{1}{N} \sum_{i=1}^{N} \int_0^{\tau_\epsilon^i} (\|\boldsymbol{u}_f^i\|^2 + \|\boldsymbol{u}_g^i\|^2)\,\mathrm{d}t$ through the Monte Carlo sampling. We show the results in Figure 4 and in Table 1 as well. Table 1 includes the training time (Tt), empirical energy cost $\mathcal{E}_{0.001}$, nearest distance (Nd) between the bicycle and target position, and empirical expectation $\mathbb{E}[\tau_{0.001}]$ for different methods. We include more experimental details in Appendix A.3.5. We can see that the ranking of the comprehensive performance is NSC > NDC > QP. This means that we can really benefit from introducing noise in the control protocol. This is reasonable because, when we regard the energy cost as an objective function for minimization, the randomness is more likely to lead this functional to the shortest path, akin to the common case where the stochastic gradient descent outperforms the full-batch gradient descent. We show the NSC can enlarge the region of attraction of the 100-D gene regulatory networks in Appendix A.4.2.

**Uncontrollable Fluctuation** The neural stochastic method we propose outperforms the control methods including the deterministic control. However, the method can cause uncontrollable fluctuation due to the stochasticity. In practice, we always want to bound this perturbation owing to physical and engineering restrictions in the real world. We tackle this safety guarantee problem in Section 5.

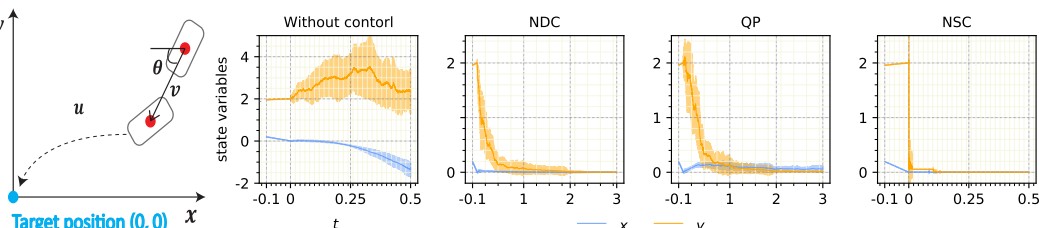

Figure 4: (Left) A schematic diagram of the kinematic bicycle model. (Right) Time trajectories of the state variables $x, y$ of the kinematic bicycle under different control cases. The solid lines are obtained through averaging the 10 sampled trajectories, while the shaded areas stand for the standard errors.

## 5 SAFETY GUARANTEE FOR SDDEs

In this section, we study the safety and stability guarantees for the SYNC framework. Based on the stochastic control barrier functions, we establish an analytical result on the safety guarantee problem for SDDEs, which guarantees that the process $\boldsymbol{x}(t;\xi)$ satisfies the safety constraint, i.e., $\boldsymbol{x}(t;\xi) \in \text{int}(\mathcal{C})$ for all $t$ with the initial value $\xi(0) \in \text{int}(\mathcal{C})$. Here, $\mathcal{C} = \{\boldsymbol{x} : h(\boldsymbol{x}) \geq 0\}$ is a compact set and the local Lipschitz function $h \colon \mathbb{R}^d \to \mathbb{R}$ is called a stochastic control barrier function (SCBF). Inspired by

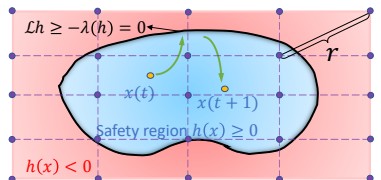

Figure 5: Diagram of the safety guarantee. We check the safety condition on discretization points with mesh $r$.

(Lechner et al., 2022), we prove that the safety and stability conditions for NN form functions can be guaranteed through a stronger condition on finite samples. We include the analytical proofs for all the results in Appendix A.1.

**Definition 5.1** *A continuous function $\alpha : (-b, +\infty) \to (-\infty, +\infty)$ is said to be of an extended class-$\mathcal{K}$ function for some $b > 0$ if it is strictly increasing and $\alpha(0) = 0$.*

**Baseline** We extend the recent results on stochastic control barrier functions in SDEs (Clark, 2019) to the SDDEs and summarize the results in Proposition 5.1. With this proposition and Theorem 2.2, the traditional deterministic control methods based on the Quadratic Program (QP) in (Fan et al., 2020; Sarkar et al., 2020) can be applied to test on the SDDEs. We use this QP method as the baseline and the specific algorithm is shown in Appendix A.3.1. We also take the classic MPC method as the baseline.

**Proposition 5.1** *Let the function $\mathcal{B} \colon \mathbb{R}^d \to \mathbb{R}$ be locally Lipschitz and twice-differentiable on $\text{int}(\mathcal{C})$. If there exist three extended class-$\mathcal{K}$ functions $\alpha_{1,2,3}(\boldsymbol{x})$ such that $[\alpha_1(h(\boldsymbol{x}))]^{-1} \leq \mathcal{B}(\boldsymbol{x}) \leq [\alpha_2(h(\boldsymbol{x}))]^{-1}$, and $\mathcal{LB}(\boldsymbol{x}, \boldsymbol{y}, t) \leq \alpha_3(h(\boldsymbol{x}))$ for the SDDE in (1). Then, $\mathbb{P}\big(\boldsymbol{x}(t) \in \text{int}(\mathcal{C})\big) = 1$ for all $t$, provided with $\boldsymbol{x}(0) \in \text{int}(\mathcal{C})$.*

A natural idea is to integrate Proposition 5.1 into our proposed neural control framework, but the main drawback in the usage of this proposition is that $\mathcal{B}(\boldsymbol{x})$ is unbounded on $\mathcal{C}$, lacking Lipschitz continuity. This drawback makes it impossible to fulfill the expected conditions only through numerical verification on finite samples. To conquer the difficulty, we propose the following theorem for safety guarantee, which, we believe, is a significant promotion of the existing barrier function theory.

**Theorem 5.2** *For the SDDE specified in (1), where $F$ and $G$ satisfy locally Lipschitz condition and locally linear growth condition, if there exists an extended class-$\mathcal{K}$ function $\lambda(x)$ such that $\mathcal{L}h \geq -\lambda \circ h$ for $\boldsymbol{x} \in \mathcal{D}$, where $\circ$ represents the function composition, $\mathcal{D}$ is compact and $\mathcal{C} \subset \mathcal{D}$. Then, the solution satisfies $\mathbb{P}(\boldsymbol{x}(t;\xi) \in \text{int}(\mathcal{C})) = 1$ for any $\xi \in C_{\mathcal{F}_0}([-\tau, 0]; \mathbb{R}^d)$ with $\xi(0) \in \text{int}(\mathcal{C})$.*

**Discretization and Safety Guarantee.** Based on the Theorem 5.2, we can construct a neural candidate class-$\mathcal{K}$ function $\lambda$ and combine it with the NDC and NSC to learn a safe controller, where the candidate $\lambda$ is required to satisfy the condition assumed in Theorem 5.2. However, the main difficulty is to guarantee the condition for every point $\boldsymbol{x} \in \mathcal{D}$, since, in practice, we can basically guarantee this condition on a finite number of training data $\tilde{\mathcal{D}}$ with $\tilde{\mathcal{D}}$ being a discretization of $\mathcal{D}$. Surprisingly, the following theorem suggests that we only need to check a slightly stronger condition on a finite number of states in $\tilde{\mathcal{D}}$ in order to establish the safety guarantee on the whole $\mathcal{D}$.

**Theorem 5.3** *Let $M = \mathcal{M}(F, G, h, \lambda, \mathcal{D})$ be the maximum of the Lipschitz constants of $\mathcal{L}h$ and $\lambda \circ h$ on $\mathcal{D}$. Also, let $r$ be the mesh size of $\tilde{\mathcal{D}}$. Thus, for each $\boldsymbol{x} \in \mathcal{D}$, there exists $\tilde{\boldsymbol{x}} \in \tilde{\mathcal{D}}$ such that $\|\boldsymbol{x} - \tilde{\boldsymbol{x}}\|_2 < r$. Suppose there exists a non-negative constant $\delta \leq Mr$ such that*

$$-\mathcal{L}h - \lambda \circ h + 4Mr \leq \delta, \; \forall \boldsymbol{x} \in \tilde{\mathcal{D}}. \tag{11}$$

*Then, $\lambda$ satisfies the safety condition specified in Theorem 5.2.*

**Remark 5.4** *Here, the non-negative $\delta$ is regarded as the tolerance error in the training stage. So, practically, we terminate the training until the safety loss is smaller than $Mr$.*

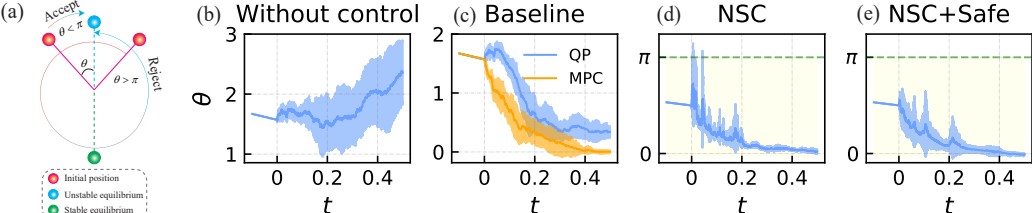

Figure 6: Schematic diagram of inverted pendulum task (a). The $\theta$ component of the original system (b), under baseline control (c), under NSC (d), and under our proposed safe control (e). The solid lines are obtained through averaging the 5 sampled trajectories, while the shaded areas stand for the standard errors.

**Construct Neural Networks with Bounded Lipschitz Constant.** We can define the loss function for safety in the manner of the left-hand side in (11). However, $M$ depends on the Lipschitz constants of the NN functions $\lambda$ and $\boldsymbol{u}$, which probably makes it complex and difficult to train the loss function. To simplify the loss function, we construct the NNs with bounded Lipschitz constants for $\lambda$ and $\boldsymbol{u}$. Specifically, we add the spectral normalization for the neural control function to constrain its Lipschitz constant lower than 1 (Miyato et al., 2018; Yoshida & Miyato, 2017). We apply the monotonic NNs to construct the candidate extended class-$\mathcal{K}$ function as $\lambda_{\boldsymbol{\theta}_\lambda}(x) = \int_0^x q_{\boldsymbol{\theta}_\lambda}(s)\mathrm{d}s$, where $q_{\boldsymbol{\theta}_\lambda}(\cdot)$, the output of the NNs, is definitely positive (Wehenkel & Louppe, 2019). To constrain the Lipschitz constant of $\lambda_{\boldsymbol{\theta}_\lambda}$, we modify the integral formula as $\lambda_{\boldsymbol{\theta}_\lambda}(x) = \int_0^x \min\{q_{\boldsymbol{\theta}_\lambda}(s), M_\lambda\}\mathrm{d}s$, where $M_\lambda$ is a predefined hyperparameter. Thus, the Lipschitz constant of $\lambda_{\boldsymbol{\theta}_\lambda}$ is smaller than $M_\lambda$. Therefore, we can calculate $M$ from the considered functions and $M_\lambda$. Other Lipschitz regularization methods can be applied in our framework (Gouk et al., 2021; Liu et al., 2022) as well.

**SYNC Algorithm:** We define the loss function for the safety guarantee of the controlled system (4) as follows, (the specific algorithms are summarized in Algorithm 1 and 2)

$$L_{\tilde{\mathcal{D}}, M_\lambda}(\boldsymbol{\theta}, \boldsymbol{\theta}_\lambda) = \frac{1}{|\tilde{\mathcal{D}}|^2} \sum_{(\boldsymbol{x}, \boldsymbol{y}) \in \tilde{\mathcal{D}} \times \tilde{\mathcal{D}}} \max\{0, -\mathcal{L}h(\boldsymbol{x}, \boldsymbol{y}) - \lambda_{\boldsymbol{\theta}_\lambda}(h(\boldsymbol{x})) + 4Mr\}. \quad (12)$$

We add this loss to equation 8 and equation 10, respectively, to separately train the NDC and NSC. To obtain the safety guarantee, we terminate the training process once $L_{\tilde{\mathcal{D}}, M_\lambda}(\boldsymbol{\theta}, \boldsymbol{\theta}_\lambda)$ is less than $Mr$.

**From Safety Guarantee to Stability Guarantee.** Akin to the safety guarantee, we provide the stability guarantee for the candidate neural control functions satisfying the condition in Theorems 2.2 and 4.1. However, both theorems require their conditions to be valid for every point $\boldsymbol{x} \in \mathcal{X} \subset \mathbb{R}^d$, while, in practice, it is impossible to obtain a finite discretization or a bounded Lipschitz constant on the unbounded $\mathcal{X}$. Ingeniously, this difficulty can be conquered with the help of the safety guarantee since the safety condition restricts $\mathcal{X} \subset \mathcal{D}$ where $\mathcal{D}$ is compact. As such, we can establish theoretical results on stability guarantee for NDC and NSC in a similar manner as that in Theorem 5.3. We thus summarize all these results in Appendix A.1.8.

We test the proposed safe control method to suppress the fluctuations emergent in the control process on the task of controlling noise-perturbed inverted pendulum with time-delay. This control task is a standard nonlinear control problem for testing different control methods (Anderson, 1989; Huang & Huang, 2000). We apply the safe control method to steer the system to the upright position without rotating a semi-circle, i.e. $|\theta| \leq \pi$. The results are shown in Figure 6 and the experimental details are provided in Appendix A.3.6. It is observed that the safe control method significantly outperforms the baseline and the stochastic control method in terms of stabilization and safety guarantee.

## 6 THEORETICAL RESULTS FOR NDC AND NSC

We have mentioned the stopping time and the energy cost in section 4.1 and numerically compare the proposed neural controllers with these indexes. These two indexes are the classic factors to measure the performance of the controller (Sun et al., 2017). From the construction in Section 5, we circumscribe the Lipschitz constant $k_{\boldsymbol{u}}$ of the control function. Based on the safety and stability guarantee, the neural controller thus satisfies the conditions assumed in Theorems 2.2 and 4.1.

Then, we have the following theoretical results and include their proofs in Appendix A.1.9.

**Theorem 6.1** *(Estimation for NDC)* Consider the SDDE with NDC controller as

$$d\boldsymbol{x}(t) = (f(\boldsymbol{x}, \boldsymbol{x}(t-\tau)) + \boldsymbol{u}_f(\boldsymbol{x}(t), \boldsymbol{x}(t-\tau))dt + g(\boldsymbol{x}(t), \boldsymbol{x}(t-\tau))dB_t, \ \ \boldsymbol{x}(0) = \boldsymbol{x}_0 \in \mathbb{R}^d,$$

where $\|f(\boldsymbol{x}, \boldsymbol{y}) - f(\bar{\boldsymbol{x}}, \bar{\boldsymbol{y}})\| \vee \|\boldsymbol{u}_f(\boldsymbol{x}, \boldsymbol{y}) - \boldsymbol{u}_f(\bar{\boldsymbol{x}}, \bar{\boldsymbol{y}})\| \le L(\|\boldsymbol{x} - \bar{\boldsymbol{x}}\| + \|\boldsymbol{y} - \bar{\boldsymbol{y}}\|)$. Assume that the controlled system satisfies the conditions assumed in Theorem 2.2 and Remark 3.1 with $\mathrm{Ker}(w_1 - w_2) = \boldsymbol{0}$. Denote by $\eta_\varepsilon = \inf\{t > 0 : \|\boldsymbol{x}(t)\| = \varepsilon\}$ the stopping time and by $\mathcal{E}(\eta_\varepsilon, T) = \mathbb{E}[\int_0^{\eta_\varepsilon \wedge T} \|\boldsymbol{u}(\boldsymbol{x}(s), \boldsymbol{x}(s-\tau))\|^2 ds]$ the corresponding energy cost in the control process with $\epsilon < \|\boldsymbol{x}_0\|$. Thus, using the same notations in Theorem 2.2, we have

$$\begin{cases} \mathbb{E}[\eta_\epsilon] \le T_\epsilon = \dfrac{V(\boldsymbol{x}_0) - \min_{\|\boldsymbol{x}\|=\varepsilon} V(\boldsymbol{x}) + \int_{-\tau}^0 w_2(\xi(s))ds}{\min_{\|\boldsymbol{x}\|\ge\varepsilon}(w_1(\boldsymbol{x}) - w_2(\boldsymbol{x}))}, \\[2ex] \mathcal{E}(\eta_\epsilon, T_\epsilon) \le \dfrac{k_{\boldsymbol{u}}^2 C_0}{2(L^2 + L + k_{\boldsymbol{u}})} \left[\exp\left(4(L^2 + L + k_{\boldsymbol{u}})T_\varepsilon\right) - 1\right] + \int_{-\tau}^0 k_{\boldsymbol{u}}^2 \xi^2(s)ds. \end{cases}$$

where $C_0 = \|\boldsymbol{x}_0\|^2 + (2L^2 + L + k_{\boldsymbol{u}})\int_{-\tau}^0 \xi(s)^2 ds$ and $\xi \in C[-\tau, 0]$ is the initial data.

We provide the similar theoretical results for NSC in Appendix A.1.10.

## 7 RELATED WORKS

**Stability Theory of SDDEs.** The early endeavors to develop the stability theory for SDDEs were attributed to (Mao, 1999; 2002) inspired by LaSalle's theory (LaSalle, 1968). The subsequent developments have been systematically and fruitfully achieved in the last twenty years in the control community Appleby (2003); Song et al. (2014); Liu et al. (2016); Zhu (2018); Peng et al. (2021). These works reveal the positive effect of multiplicative noise to the stochastic dynamics with delays, and motivate us to develop *only* neural stochastic control to stabilize dynamical systems.

**Finding Stabilization Controller.** Traditional control methods focus on transforming control criteria, such as the control Lyapunov functions (CLFs), into the QP (Fan et al., 2020; Sarkar et al., 2020) or the semi-definite planning (SDP) problems (Henrion & Garulli, 2005; Jarvis-Wloszek et al., 2003; Parrilo, 2000) to find optimal control iteratively. These methods have high computational complexity since they cannot give the closed form of the control. Hence, machine-learning-based control methods have been introduced to improve the generalization and efficiency of the original convex optimal problems (Khansari-Zadeh & Billard, 2014; Ravanbakhsh & Sankaranarayanan, 2019; Gurriet et al., 2018). However, all the existing learning methods consider dynamics without time-delay (Wagener et al., 2019; Williams et al., 2018; Chang et al., 2019; Zhang et al., 2022).

**Theory and Application of Control Barrier Function** The barrier function method has been extensively researched in the problem of safety verification of controlled dynamics (Prajna & Jadbabaie, 2004; Jankovic, 2018; Prajna et al., 2004; Clark, 2019; 2021). Existing works for constructing barrier functions in applications typically based on quadratic programming (Ames et al., 2014; 2016; Khojasteh et al., 2020; Fan et al., 2020). Machine learning methods have also been introduced in safe control fields in (Robey et al., 2020; Dean et al., 2020; Taylor et al., 2020).

## 8 DISCUSSION

We heuristically design two kinds of neural controllers for SDDEs based on the classic LaSalle-type stabilization theory and the newly proposed stochastic stabilization theorem. To assure the controlled trajectories can stay in the safety region, we cultivate the safety guarantee theorem through the SCBF and the discretization techniques. Since the state space of the controlled SDDEs with safety guarantee is bounded by the compact safety region, we can similarly deduce the stability guarantee theorem for neural controllers through spatial discretization. Furthermore, we theoretically and numerically investigate the neural controllers' performance in terms of convergence time and energy cost. The proposed neural controllers with safety and stability guarantee are summarized as SYNC, which significantly simplifies the process of control design and has extensive potential in different control fields, such as financial engineering (Zhou & Li, 2000).

## 9 ACKNOWLEDGMENTS

We thank the anonymous reviewers for their valuable and constructive comments that helped us to improve the work. Q.Z is supported by the China Postdoctoral Science Foundation (No. 2022M720817), by the Shanghai Postdoctoral Excellence Program (No. 2021091), and by the STCSM (Nos. 21511100200 and 22ZR1407300). W.Y. is supported by the STCSM (Nos. 21511100200, 22ZR1407300 and 22dz1200502). W.L. is supported by the National Natural Science Foundation of China (No. 11925103) and by the STCSM (Nos. 22JC1402500, 22JC1401402, and 2021SHZDZX0103).

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
