# OpenReview forum: "SYNC: SAFETY-AWARE NEURAL CONTROL FOR STABILIZING STOCHASTIC DELAY-DIFFERENTIAL EQUATIONS"
_ICLR.cc/2023/Conference — ICLR 2023 poster_

### Official Review · Reviewer_HvjV · 2022-10-21

**Confidence:** 2
**Clarity, Quality, Novelty And Reproducibility:** As mentioned in the weaknesses sectio…
**Correctness:** 3
**Technical Novelty And Significance:** 2
**Empirical Novelty And Significance:** 2
**Recommendation:** 8

**Strength And Weaknesses:**

# Strengths:
- The paper relies on minimal assumptions (twice continuously differentiable systems and policies).
- The paper introduces a method for an important problem (stable and safe control in stochastic systems)

# Weaknesses:
Generally, I feel the writing of the paper could be a bit clearer. For instance, the function in Eq (5) looks quite similar to the neural network learned in Chang et al. 2019.
Moreover, the failure case of the NDC is not entirely clear to me. How exactly does the limit of x at infinity cause the limitation of the NDC?
What is the role of the L2 regularization in the method and the flaw of NDC specifically?
Overall, the structure of the paper is good though.

The contributions seem rather incremental. In particular, the difference to Chang et al. 2019 seems to be only in the time-delay of the model. The main novelty of the paper compared to stochastic control barrier functions is mentioned to be its simplicity. Both arguments seem rather minor advancements.

The scale of the experiment seems rather limited. Particularly, the bicycle model and the inverted pendulum task are both use cases that much simpler linear or LQR control policies can potentially solve.
This raises the question of what bottlenecks the method.

How does the LaSalle's type loss ensure the fulfillment of the conditions of Theorem 2.2 for all (x,y), especially when the function V is expressed by an MLP and trained by gradient descent on random samples?

To obtain safety guarantees, the paper uses an input convex neural network. I don't see how this is then more expressive than a QP-based approach to synthesizing control barrier functions.
Can you elaborate?


**Summary Of The Paper:**

The paper proposed a method for generating stable and safe policies for stochastic delay-differential equations. Specifically, the paper achieves this by learning a LaSalle's type stability certificate as an MLP and extending it to a stochastic control barrier function.

**Summary Of The Review:**

Overall interesting work simplifying existing approaches model and adding time-delay to the model. Contributions compared to existing methods may be incremental and experimental evaluation is limited to simpler control tasks.

---

> ### Author Response · Authors · 2022-11-12
> **Response 1/2**
>
> We thank you for your overall positive feedback, valuable comments, and helpful suggestions on the proposed methods. For your minor/major points in this work, we respond to them one by one. Hopefully, the following answers could help you understand our work better.
>
> ---
>
> > **Q1**: The function in Eq (5) looks quite similar to the neural network learned in [Chang et al. 2019].  What is the role of the L2 regularization in the method?
>
> **Response**: Many thanks for your comment. The auxiliary function $V$ in [Chang et al. 2019] is a Lyapunov function for ODEs, satisfying the Lyapunov conditions in a bounded region. In our paper, we nontrivially generalize their work to the case of SDDEs. In fact, an introduction of stochasticity and time-delay definitely brings more difficulties in constructing the function $V$.  Moreover, we use the L2 regularization to guarantee the conditions for $\lim_{\|x\|\to\infty}V(x)=\infty$.
>
> ---
>
> > **Q2**: How exactly does the limit of $x$ at infinity cause the limitation of the NDC?
>
> **Response**: Thanks for your valuable comments. The failure of NDC in stochastic control case is caused by $l(\theta_g)\ge0$ on some samples, where $\theta_g$ indicates a group of parameters awaiting estimation by training. Minimizing the loss is equivalent to minimizing the $l(\theta_g)\ge0$. The best result after training is $l(\theta_g)=0$, so that $u_g=0$. This means that the best result implies a zero stochastic control. However, this zero stochastic control cannot stabilize the original unstable system. To avoid $l(\theta_g)\ge0$ on training data, the $V$ must be designed such that its maximum eigenvalue $\lambda_\max(\mathcal{H}V)\le-\delta$ for some positive $\delta$, which contradicts $\lim_{\|x\|\to\infty}V(x)=\infty$.
>
> ---
>
> > **Q3**: What is the flaw of NDC specifically?
>
> **Response**: Thanks for this question.  Although the NDC outperforms the existing linear control method for SDDEs, there are two major flaws of NDC. One is the failure in the stochastic control case as we explained above. The other is the high computational complexity due to the need for calculating a hessian matrix in $\mathcal{L}V$. Specifically, the computational complexity for NDC is $\mathcal{O}((mn)^2)$ for ${\rm{batch~size}}=m$ on $n$-D dynamics while the computational complexity for NSC is $\mathcal{O}(mn)$. We further discuss these results in Appendix A.3.2.
>
> ---
>
> > **Q4**:  The difference to [Chang et al. 2019] seems to be only in the time-delay of the model. The main novelty of the paper compared to stochastic control barrier functions is mentioned to be its simplicity. Both arguments seem rather minor advancements.
>
> **Response**: Thanks for your suggestion of emphasizing the difference with the existing works. The difference between NDC and [Chang et al. 2019] is the formulation of the problem, we tackle the stochastic dynamics with time-delay, while [Chang et al. 2019] only consider the dynamics without stochasticity and time-delay. The stochasticity and time-delay significantly increase the difficulty of controlling dynamics. We also provide the safety and stability guarantee for neural controllers, which is not solved in [Chang et al. 2019]. The Lipschitz continuous SCBF in Theorem 5.2 allows us to obtain the safety guarantee analytically. But the original SCBF $\mathcal{B}$ in Proposition 5.1 is unbounded on safety region $\mathcal{C}$ and lacks Lipschitz continuity, which makes it impossible to obtain the safety guarantee in neural network settings. Thus, we make major improvements compared to the existing works.
>
> ---
>
> > **Q5**: Particularly, the bicycle model and the inverted pendulum task are both use cases that much simpler linear or LQR control policies can potentially solve. This raises the question of what bottlenecks the method.
>
> **Response**: Thanks for your valuable comments. In the revision, we focus on providing the safety guarantee and stability guarantee for controlled SDDEs in neural network settings. To our best knowledge, there lack the similar guarantees of linear control or LQR control in SDDEs. We also supplement an experiment of controlling 100-D gene regulatory networks in Appendix A.4.2 to demonstrate the performance of the proposed neural controllers.

---

> > ### Author Response · Authors · 2022-11-12
> > **Response 2/2**
> >
> > > **Q6**:  How does the LaSalle's type loss ensure the fulfillment of the conditions of Theorem 2.2 for all (x,y), especially when the function V is expressed by an MLP and trained by gradient descent on random samples?
> >
> > **Response**: Thanks for your suggestion. In the revision, we first revise the safety condition with Lipschitz SCBF on safety region $\mathcal{C}$ in Theorem 5.2. Based on spatial discretization and Lipschitz continuity, we supplement Theorem 5.3 to show it is sufficient to check a slightly stricter condition on finite samples to ensure the fulfillment of the condition in Theorem 5.2. Since the safety guarantee limits the state space in a bounded safety region, we can similarly guarantee the fulfillment of conditions of Theorem 2.2 and Theorem 4.1 for all states in the safety region. We summarized the results in Appendix A.1.8.
> >
> > ---
> >
> > > **Q7**: To obtain safety guarantees, the paper uses an input convex neural network. I don't see how this is then more expressive than a QP-based approach to synthesizing control barrier functions. Can you elaborate?
> >
> > **Response**: Thanks for your careful reading and questions. We replace the input convex neural network with an unconstrained monotonic neural network (UMNN) to obtain the safety guarantee. The neural network provides an analytical form of SCBF, while the QP method solves an optimal problem for each state to obtain function value at the state. Moreover, in our numerical simulation, we extend the existing QP-based method to SDDEs and compare it with our method, the results show that our method outperforms the QP-based method. In practice, the QP-based approach fixed the function form of SCBF and focus on finding the control function [1,2].
> >
> > ---
> >
> > We thank the reviewer again for your valuable comments. We do believe that the quality of the revised paper has been improved exceptionally from the experimental and theoretical aspects thanks to the reviewers’ comments and suggestions. Hopefully, the responses and the revised paper have sufficiently addressed the main concerns and then the reviewer will reconsider the assessment in support of the revised paper. We are looking forward to your feedback to make further improvements to the paper.
> >
> > [1] Sarkar, M., Ghose, D., & Theodorou, E. A. (2020). High-relative degree stochastic control lyapunov and barrier functions. arXiv preprint arXiv:2004.03856.
> >
> > [2] Fan, D. D., Nguyen, J., Thakker, R., Alatur, N., Agha-mohammadi, A. A., & Theodorou, E. A. (2020, May). Bayesian learning-based adaptive control for safety critical systems. In 2020 IEEE international conference on robotics and automation (ICRA) (pp. 4093-4099). IEEE.

---

> > > ### Comment · Reviewer_HvjV · 2022-11-22
> > > **Thanks for the revision**
> > >
> > > The authors have answered my questions clearly and revised their work based on my comments.
> > >
> > > I have updated my score accordingly.

---

> > > > ### Author Response · Authors · 2022-11-22
> > > > **Thank you very much**
> > > >
> > > > We thank you very much for your comments and your positive support to our works. Also, we are happy to hear that many of your concerns are addressed.
> > > >
> > > > Best,
> > > >
> > > > Authors

---

### Official Review · Reviewer_Tymv · 2022-10-30

**Confidence:** 2
**Correctness:** 3
**Technical Novelty And Significance:** 4
**Empirical Novelty And Significance:** Not applicable
**Recommendation:** 6

**Clarity, Quality, Novelty And Reproducibility:**

The work seems novel (though this paper is outside my area, so I'm not able to fully evaluate this). The quality seems reasonable except for the over-claim of guaranteed stability, and my uncertainty regarding whether the problem formulation itself is standard or novel (if the latter, it needs to be further justified). The authors provide code to improve reproducibility, and additional details are given in the appendix. The clarity could be significantly improved.

**Strength And Weaknesses:**

Strengths:
* The amount of theoretical and empirical work represented within this submission is seemingly immense. The authors derive new stability theorems, and structure neural network controllers that play well with the assumptions associated with both existing theorems and these new theorems. The authors also run numerous experiments to demonstrate the efficacy of their framework.

Weaknesses:
* While the authors claim they obtain a safety guarantee through their neural network control framework, as far as I can tell, this is not the case. In particular, while some assumptions within the theorems they build upon are satisfied through the structure of their neural network framework, other assumptions and preconditions are only _incentivized_ (not guaranteed) to be satisfied through the design of loss functions. As such, I don't think the authors can actually claim that their controller "guarantees" safety.
* In part because of the immense quantity of the work in the submission, the paper as written is rather inaccessible for an ICLR audience and is not self-contained. In particular, the authors could spend much more time describing the intuition behind their approach, providing algorithm diagrams and flow charts, etc. in the main text to really impart the big ideas; right now, the main body of the paper is hard to follow without this clearly articulated conceptual framework. The experimental results are also not self-contained within the main paper.
* The mathematical notation is not adequately defined, given that some of it is not standard or usual for the ICLR audience.

Questions:
* Is the problem statement, and particularly the formulation in Equation (4), standard - or is it fully newly proposed by the authors? If the latter, what are the strengths and limitations of the proposed formulation itself (in addition to the solution technique)?

**Summary Of The Paper:**

This paper provides a safety-aware control approach, based on neural networks, for stabilizing stochastic delay differential equations (SDDEs). In particular, the authors construct a controller with two components - a deterministic control $u_f$ tied to the time-evolution of the system, and a stochastic control $u_g$ tied to the Brownian motion of the system. The authors both build off existing theorems and derive new theorems to characterize conditions under which the controllers will stabilize the system. They then construct their neural networks to satisfy some of the assumptions within these theorems, as well as constructing loss functions that incentivize satisfaction of the rest. The authors demonstrate the superiority of their method compared to baselines on small-scale experiments.

**Summary Of The Review:**

The authors provide a seemingly novel, interesting framework for constructing neural network controllers for SDDEs. However, the claim that this framework "guarantees" safety is seemingly an overclaim, and the submission needs to be significantly improved in terms of its clarity and accessibility for the ICLR audience. In addition, the authors should further clarify where their problem formulation in Equation (4) comes from, and further justify it if it is a novel problem formulation.

---

> ### Author Response · Authors · 2022-11-12
> **Response**
>
> We would like to thank the reviewer for the overall positive feedback and helpful suggestions. We revised the paper carefully according to the reviewer’s comments. Our major changes in the revised paper are listed in “Response to all reviewers”. We address all the major concerns of the reviewer point by point as follows.
>
> ---
>
> > **Q1**: Some assumptions and preconditions are only incentivized (not guaranteed) to be satisfied through the design of loss functions.
>
> **Response**: Thanks for your careful reading and helpful suggestion. In the revision, we provide
>  analytical results to show that we can guarantee the expected conditions. Specifically, in Theorem 5.2, we provide a new safety condition, replacing the unbounded SCBF in the original theorem with a bounded SCBF on safety region $\mathcal{C}$. Then, based on the spatial discretization and Lipschitz continuity, we show in Theorem 5.3 that it is sufficient to check a slightly stricter condition on finite discretized states to establish the safety guarantee on the whole $\mathcal{C}$. We construct neural networks with uniform Lipschitz constant based on the spectral regularization and the unconsrtained monotonic neural network (UMNN) to limit the Lipschitz constant in Theorem 5.3. Similarly to the safety guarantee, we further obtain the stability guarantee for NDC and NSC, and we summarize them in Appendix A.1.8.
>
>
> ---
>
> > **Q2**: The main body of the paper is hard to follow without this clearly articulated conceptual framework.
>
> **Response**: Thanks for your comment. We provide more explanations for illustrating Theorem 4.1 and the theorems in Section 5. Also, we depict a diagram for the safety guarantee in Section 5. Due to the page limit, we put the algorithms in Appendix A.3.1.
>
> ---
>
> > **Q3**: The mathematical notation is not adequately defined, given that some of it are not standard or usual for the ICLR audience.
>
> **Response**: Thanks for your helpful suggestion. We provide detailed descriptions for all the mathematical notations in Appendix A.1.1.
>
> ---
>
> > **Q4**:  Is the problem statement, and particularly the formulation in Equation (4), standard - or is it fully newly proposed by the authors? If the latter, what are the strengths and limitations of the proposed formulation itself (in addition to the solution technique)?
>
> **Response**:  Many thanks for your comment. The standard stochastic control problems focus on the controlled equations as $\mathrm{d}x=(f+u_f)\mathrm{d}t+g\mathrm{d}B_t$ [1], where the controller is only emergent in the drift terms. However, we include the controller in the diffusion term, making it possible to investigate the positive role of noise in controlling SDDEs. Through the numerical simulations, we find that a stochastic controller can stabilize the system faster than a deterministic controller. We include more discussion on the difference of the stochastic control and the deterministic control in Appendix A.3.2.
>
> ---
>
> [1] Clark, A. (2019, July). Control barrier functions for complete and incomplete information stochastic systems. In 2019 American Control Conference (ACC) (pp. 2928-2935). IEEE.

---

> > ### Author Response · Authors · 2022-11-25
> > **Sincerely Looking Forward to Your Reply**
> >
> > Dear Reviewer Tymv,
> >
> > Thanks again for your time and efforts in reviewing our paper. Your valuable comments and suggestions do help us polish the paper, from the safety guarantee (**Q1**), the readability of the paper (**Q2**, **Q3**), and the problem statement (**Q4**).
> >
> > We are sincerely looking forward to your reply, and we hope you will consider increasing your score if we have addressed your concerns.
> >
> > Best,
> >
> > Authors

---

> > > ### Comment · Reviewer_Tymv · 2022-11-28
> > > **Response**
> > >
> > > Thanks to the authors for their thoughtful revisions, and my sincere apologies for the delayed reply.
> > >
> > > Unfortunately, my major concerns still stand:
> > >
> > > 1. It is possibly due to my own lack of understanding, but I'm not able to follow how Section 5 fits in with the theory and methods proposed in Sections 3 and 4. In addition, Section 5 still seems to be based on incentivizing safety through a loss function, rather than guaranteeing it (for instance, the authors state that "we can terminate the training process once $L_{\tilde{D}, M_\lambda}(\theta, \theta_\lambda)$ is less than $Mr$" but don't show that this is guaranteed to occur).
> > >
> > > 2. In general, I still think much more needs to be done to make this submission accessible to the ICLR audience. While I appreciate the additional explanations the authors have added, in my opinion, the text requires a significant overhaul to make the "big picture" story clearer - i.e., how different parts of the theory/method fit together. The authors should also consider adding a diagram illustrating this.
> > >
> > > 3. Regarding the problem formulation, thank you for the clarification. Since the authors are proposing a new formulation incorporating the controller in the diffusion term, it would be important to motivate in practice why (beyond mathematical convenience) this can/should be done. Are there real-world scenarios in which this kind of control is possible/why has this formulation not been proposed before?
> > >
> > > Given that my major concerns still stand, I am currently keeping my score the same. It is very possible that concerns #1 and 3 are addressed, and that I am missing something - that said, either way, I do think clarity would need to be significantly improved (concern #2 above) for this paper to be published at ICLR.

---

> > > > ### Author Response · Authors · 2022-11-28
> > > > **Further Response 1/2**
> > > >
> > > > We would like to thank the reviewer again for the careful reading and helpful suggestions. We revised the paper carefully according to the reviewer’s comments. We address all the major concerns of the reviewer point by point as follows. Since the deadline for updating the paper has passed, we put the revision on the anonymous GitHub (https://anonymous.4open.science/r/SYNC-9528/1129_full.pdf), and we will update the camera-ready paper if our work could be accepted.
> > > >
> > > > ---
> > > >
> > > > __Response to concern 1__:  Thanks for your valuable comments. In Section 5, we first establish the safety guarantee for the neural controller in Theorem 5.2, 5.3. Specifically, once the safety-aware loss function satisfies the condition in Eq.(11) on the training data, the controlled trajectories will stay in the bounded safety region $\mathcal{C}$. Based on this result, we can further obtain the stability guarantee for the neural controller in Sections 3 and 4 due to the boundness of the state space $\mathcal{X}\subset\mathcal{C}$. We summarize the analytical results in Appendix A.1.8. We note that the conclusion in Theorem 5.3 still stands with the stronger condition $-\mathcal{L}h-\lambda\circ h+3Mr\le 0$, $\forall x \in \tilde{\mathcal{D}}$. However, in the training process, the loss does not necessarily converge to zero due to the data and model errors. Therefore, we relax the above condition to a weaker condition $-\mathcal{L}h-\lambda\circ h+4Mr\le \delta,~\delta<Mr$. In this way, we can regard the $\delta$ as a tolerance error that we can break the training stage even if the loss is not reduced to zero. A similar operation has been used in the existing works [1-2].
> > > >
> > > > ---
> > > >
> > > > __Response to concern 2__:   Many thanks for your valuable suggestion. To make the paper more readable, we redraw the diagram in Figure 1 to illustrate the connections between the proposed theorems and the sections.  In addition, we make an overhaul to the paper to explain the whole work more clearer, including the discussion in Section 8 and the opening parts in Section 3,4,5,6. Here, to help ICLR audiences understand how different parts of the theory/method fit together, we briefly introduce the main idea of our work. First, we heuristically design two kinds of neural controllers for SDDEs based on the classic LaSalle-type stabilization Theorem 2.2 and the newly proposed stochastic stabilization Theorem 4.1. To assure the controlled trajectories can stay in the safety region, we cultivate the safety guarantee Theorem 5.2&5.3 through the SCBF and the discretization techniques. Since the state space of the controlled SDDEs with safety guarantee is bounded by the compact safety region, we can similarly deduce the stability guarantee theorem for neural controllers through spatial discretization. Moreover, we theoretically and numerically investigate the neural controllers' performance in terms of convergence time and energy cost.
> > > >
> > > > We hope that the revised article as well as the responses adequately addresses the reviewers’ concerns, and we are more than happy to address your further questions for this paper to make the work more readable and valuable.
> > > >
> > > > ---
> > > >
> > > > __Response to concern 3__: Many thanks for your careful reading and helpful suggestion. The formulation of the stochastic controller in the diffusion term has been proposed in stochastic linear quadratic control problems (see [3-4]). The research on this problem has significantly impacted financial engineering applications, such as portfolio design [5]. Time delay has also been considered for this kind of LQ problem [6]. However, existing works mainly focus on the linear equation case due to the complexity brought by stochasticity. Moreover, the optimal controller for nonlinear dynamics is, in general, intractable. In our framework, we utilize neural networks to approximate the optimal controller that can stabilize the nonlinear dynamics of the equlibrium with safety and stability guarantees.

---

> > > > > ### Author Response · Authors · 2022-11-29
> > > > > **Further Response 2/2**
> > > > >
> > > > > **References**
> > > > >
> > > > > [1] Lechner, M., Žikelić, Đ., Chatterjee, K., & Henzinger, T. A. (2022, June). Stability verification in stochastic control systems via neural network supermartingales. In Proceedings of the AAAI Conference on Artificial Intelligence (Vol. 36, No. 7, pp. 7326-7336).
> > > > >
> > > > > [2] Chang, Y. C., Roohi, N., & Gao, S. (2019). Neural lyapunov control. Advances in neural information processing systems, 32.
> > > > >
> > > > > [3] Chen, S., Li, X., & Zhou, X. Y. (1998). Stochastic linear quadratic regulators with indefinite control weight costs. SIAM Journal on Control and Optimization, 36(5), 1685-1702.
> > > > >
> > > > > [4] Li, X., Zhou, X. Y., & Ait Rami, M. (2003). Indefinite stochastic linear quadratic control with Markovian jumps in infinite time horizon. Journal of Global Optimization, 27(2), 149-175.
> > > > >
> > > > > [5] Zhou, X. Y., & Li, D. (2000). Continuous-time mean-variance portfolio selection: A stochastic LQ framework. Applied Mathematics and Optimization, 42(1), 19-33.
> > > > >
> > > > > [6] Zhang, H., Li, L., Xu, J., & Fu, M. (2015). Linear quadratic regulation and stabilization of discrete-time systems with delay and multiplicative noise. IEEE Transactions on Automatic Control, 60(10), 2599-2613.
> > > > >
> > > > > ---
> > > > >
> > > > > Thanks again for your time and efforts in reviewing our paper. Your valuable comments and suggestions do help us improve the paper. We have tried our best to make the paper more readable in the revised paper, and we are more than happy to address your further concerns to improve this work. We are sincerely looking forward to your reply. Finally,  we appreciate it very much if the reviewer could find the further contributions and our efforts on this revised work compared with the original version, and then consider revising the assessment in support of the revised paper.

---

> > > > > ### Comment · Reviewer_Tymv · 2022-11-29
> > > > > **Response to Further Response**
> > > > >
> > > > > Thanks to the authors for their revisions.
> > > > >
> > > > > **Concern 1:** Thanks for the explanation. However, I still do not understand how the loss function is used, and how it relates to the construction of the NSC in Section 4. Is it added to the losses in Equations 8/10? Or are the NDC/NSC first obtained and somehow then modified with the loss?
> > > > >
> > > > > **Concern 2:** Thanks for adding the diagram, which is extremely helpful. Some remaining questions: Why can the NDC and NSC be obtained separately? Is the SYNC workflow in Section 5 acting on both of these controllers jointly?
> > > > >
> > > > > In general, I do still think there are major clarity improvements needed, which I cannot fully list out. Some illustrative examples:
> > > > > * The notation $\|\|x\|\| \vee \|\|y\|\|$ as used is, I believe, not standard (I was not able to understand what it meant)
> > > > > *  Section 4 states: "The intuition here is to construct the stability condition that makes the diffusion term contribute a negative number, contrary to the Theorem 2.2." What is being contradicted in Theorem 2.2? How does that connect to the diffusion term being negative? (Theorem 2.2 has enough notation that it is difficult to parse this out, since an intuitive explanation for what Theorem 2.2 means is not given - in general, I think intuitive explanations are needed at the theorem level, not just the section level).
> > > > >
> > > > > **Concern 3:** Thanks for providing the clarification. I no longer have a concern about the problem formulation.

---

> > > > > > ### Author Response · Authors · 2022-11-29
> > > > > > **Response to Reviewer's Further Response**
> > > > > >
> > > > > > Thank you for your further insightful feedback. Below is our response to your concerns, and the latest revision could be found on the anonymous GitHub (see https://anonymous.4open.science/r/SYNC-44D4/1129_full_v1.pdf).
> > > > > >
> > > > > > ---
> > > > > >
> > > > > > **Response to concern 1**: Many thanks for your further comments. To help readers understand how the loss function is used, and how it relates to the construction of the NSC in Section 4,  we make a further brief explanation of how the SYNC framework works in practice (please see the subsection **SYNC algorithm** in Section 5, and more details about algorithms could be found in Appendix A.3.1). Moreover, the safety loss Eq.(12) is added to the losses in Eq.(8) and Eq.(10), respectively, to train the NDC and NSC. To obtain the safety guarantee, we terminate the training process once the safety loss Eq.(12) is less than the tolerance error.
> > > > > >
> > > > > > ---
> > > > > >
> > > > > > **Response to concern 2**: Many thanks for your careful reading and helpful suggestions. We design the NDC and NSC frameworks, respectively, based on Theorem 2.2 and Theorem 4.1. These two frameworks can be obtained separately because we utilize different stability conditions in Theorem 2.2/Theorem 4.1 to construct neural networks and loss functions. As for the safety guarantee, the SYNC acts on NDC and NSC separately. Take NSC for an example, the SYNC adds safety loss Eq.(12) to stability loss Eq.(10), then trains the parameters in NSC with the summed loss.
> > > > > >
> > > > > > - To make the paper more readable, we provide the detailed notations in Appendix A.1.1. Here, for two real numbers $x,y$, $x\wedge y$ denotes the minimum of them, and $x\vee y$ denotes the maximum of them.
> > > > > >
> > > > > > - To compare the influence of diffusion term $G$ in Theorem 2.2 and Theorem 4.1, we list the sufficient conditions respectively:
> > > > > >
> > > > > > **Condition 1 (Theorem 2.2)**
> > > > > >
> > > > > > $V_t(x,t)+\nabla V({x},t)^\top F(x,y,t)+\dfrac{1}{2}\mathrm{Tr}\left[G^\top(x,y,t)\mathcal{H}V({x},t)G(x,y,t)\right]\le\gamma(t)+w_1(x)-w_2(y),$
> > > > > >
> > > > > >
> > > > > > **Condition 2 (Theorem 4.1)**
> > > > > >
> > > > > > $\|x_t(0)\|^2(2\langle x_t(0),F(x_t,t)\rangle+\|G(x_t,t)\|_{\rm F}^2 )-(2-\alpha)\|{x_t(0)}^{\top}G(x_t,t)\|^2\le 0,~\alpha\in(0,1).$
> > > > > >
> > > > > > We can see that $G$ contributes quadratic form term $\frac{1}{2}\mathrm{Tr}[G^\top(x,y,t)\mathcal{H}V({x},t)G(x,y,t)]$ in **Condition 1**. As explained in Section 3, this term is positive most of the time, which leads to the failure in finding stochastic control in this framework. To overcome this problem, we aim at designing stability conditions such that the part related to $G$ can be negative most of the time. In **Condition 2**, it can be verified that
> > > > > >
> > > > > > $\|x_t(0)\|^2(\|G(x_t,t)\|_{\rm F}^2 )-(2-\alpha)\|{x_t(0)}^{\top}G(x_t,t)\|^2$
> > > > > >
> > > > > > $\le\|x_t(0)\|^2(\|G(x_t,t)\|_{\rm F}^2(1-(2-\alpha)\cos^2(w))$
> > > > > >
> > > > > > where $w$ represent the angle between the vector $x_t(0)$ and $G(x_t,t)$. When $w$ is near $k\pi,k\in\mathbb{Z}$, we have $(1-(2-\alpha)\cos^2(w))\le0$. Hence, we can find a stochastic controller that makes the angle $w$ near to $k\pi$ to make the term $~\|x_t(0)\|^2(\|G(x_t,t)\|_{\rm F}^2 )-(2-\alpha)\|{x_t(0)}^{\top}G(x_t,t)\|^2$ less than zero.  We further add intuitive explanations for the theorems to make the paper more readable in Appendix A.1.4.
> > > > > >
> > > > > > ---
> > > > > >
> > > > > > Thanks again for your further detailed and valuable comments. The paper is polished according to your suggestions. We have tried our best to make the paper clearer, and we are looking forward to your further reply.  We would appreciate it very much if you could support our work and we are willing to make further improvements with any constructive feedback from your side.
> > > > > >
> > > > > > Thanks for your time and efforts.

---

> > > > > > > ### Comment · Reviewer_Tymv · 2022-12-09
> > > > > > > **Response to Response to Reviewer's Further Response  :)**
> > > > > > >
> > > > > > > Thanks to the authors for their revisions. I think the submission is now much clearer, and I have raised my score accordingly.

---

> > > > > > > > ### Author Response · Authors · 2022-12-09
> > > > > > > > **Thank you**
> > > > > > > >
> > > > > > > > We sincerely thank you for increasing the score, and your time and efforts.
> > > > > > > >
> > > > > > > > Best,
> > > > > > > >
> > > > > > > > Authors

---

> > > > > > ### Author Response · Authors · 2022-12-02
> > > > > > **Dear Reviewer Tymv: Are the concerns addressed and any other concerns?**
> > > > > >
> > > > > > Dear Reviewer Tymv,
> > > > > >
> > > > > > Thanks again for your comments and suggestions. Your constructive feedback helps us to adequately revise the paper and improve its quality. This is a friendly reminder that the final stage of the discussion period is drawing to a close. We would appreciate it if you could let us know if your concerns have been addressed and if we can clarify any other concerns.
> > > > > >
> > > > > > Nevertheless, Thank you for your time and kind efforts.
> > > > > >
> > > > > > Best,
> > > > > >
> > > > > > Authors

---

> > ### Author Response · Authors · 2022-12-09
> > **Gentle Reminder: 3 days left in final discussion phase**
> >
> > Dear Reviewer Tymv,
> >
> > Hope you are well! We would like to thank you for your constructive and insightful comments as well as your time and efforts.
> >
> > This is a gentle reminder that there are 3 days remaining in the final discussion phase. We are sincerely looking forward to your reply, and we are more than happy to continue the discussion during the remaining discussion period.
> >
> > Best,
> >
> > Authors

---

### Official Review · Reviewer_cu1i · 2022-11-03

**Confidence:** 4
**Correctness:** 2
**Technical Novelty And Significance:** 2
**Empirical Novelty And Significance:** 3
**Recommendation:** 6

**Clarity, Quality, Novelty And Reproducibility:**

The paper is written reasonably clear, and it contains some valuable novel results. But I have some major concerns about the authors' claim regarding theoretical stability guarantee of the closed-loop system using their proposed method, and I think that's a over-claim.



**Strength And Weaknesses:**

Strength:

[1] The problem of control of nonlinear stochastic delay-differential equations is very interesting and also extremely challenging

[2] The proposed methodology and main results were presented/explained reasonably clear, and overall speaking I found the manuscript friendly to readers.

[3] The proposed framework is novel, and the provided case studies are helpful to demonstrate the advantages and efficacy of the proposed neural-based method in comparison with the typical/traditional existing control methods.

Weakness:

[1] It was claimed in the paper abstract that the proposed NSC and NDC methods "guaranteeing the stochastic stability in SDDEs". However, based on my understanding, such proposed methods for training neural network Lyapunov functions for control synthesis of nonlinear systems are heuristic in nature, and extremely hard to obtain rigorous stability or safety guarantees for the closed-loop system. For the main theoretical results in Theorem 5.1, it was mentioned that "Assume the controlled system satisfies the similar conditions in Theorem 2.2 and Remark 3.1 with Ker(w1 − w2) = 0", but it was not clear to me how the authors guarantee that the closed-loop control system would indeed satisfy all the conditions in Theorem 2.2. Similar for other main theoretical results (e.g., Theorem 5.2). Simply adding stability conditions and safety conditions to the loss function of the NNs is not sufficient to guarantee the closed-loop controlled system would be stabilized, since there are a lot of different things affect the NN training and the NN training might even possibly diverge, and there is no much rigorous theoretical guarantee for such NN method. So I'm not convinced by the authors' claim about the proposed methods  "guaranteeing the stochastic stability in SDDEs" from the current version manuscript. More clarifications/elaborations from the authors are required to justify such claims. This is the biggest limitation/concern in my mind for the current version manuscript.

[2] It is not comprehensively discussed/explained when to use NSC and when to NDC. For the specific case study in Section 4.1, I saw a brief statement "We can see that the ranking of the comprehensive performance is that NSC is greater than NDC than QP". But is such conclusion generic for all cases, or it is just an observation for this special case study? I.e., is that using NSC rather than NDC always the recommended method by the authors (when there are randomness there), and what are the pros and cons? It would be helpful for the authors to elaborate more and share more comprehensive discussions for generic cases.


**Summary Of The Paper:**

In this paper, the authors studied the stabilization problem of the systems described by stochastic delay-differential equations, and the main contribution is the designed framework of neural deterministic and stochastic control with the analysis of stability and safety. Experiment results are given to compare the performance of  the proposed NSC/NDC method and baseline methods.

**Summary Of The Review:**

This paper proposed a novel framework for neural deterministic and stochastic control with the analysis of stability and safety, for systems described by stochastic delay-differential equations. The empirical cases studies are helpful to demonstrate the effectiveness of the proposed method.

However, my main concern about the current version manuscript is that, I'm not convinced by the authors' claim that the proposed NSC and NDC methods "guaranteeing the stochastic stability in SDDEs", and I think the proposed methods are heuristic in nature and hard to have rigorous stability or safety guarantees for the closed-loop system. Some more clarifications/elaborations about the theoretical theorems (5.1, 5.2, etc.) for the closed-loop controlled systems would be needed to justify the authors' claim about the theoretical stability guarantee of the closed-loop system.

If the authors can't rigorously prove theoretical stability guarantee of the closed-loop system, the proposed method and the empirical case studies are still somewhat valuable, though the contribution of the paper would be less significant.  In that case, it would be a boarder-line paper (slightly below acceptance threshold), and at the very least, the over-claim about the theoretical stability guarantee needs to be corrected/clarified.

---

> ### Author Response · Authors · 2022-11-12
> **Response**
>
> We thank the reviewer for the valuable comments. We address all the major concerns point by point as follows.
>
> ---
>
> > **Q1**:  How do the authors guarantee that the closed-loop control system would indeed satisfy all the conditions in the main Theorems?
>
> **Response**: Many thanks for your insightful suggestion. In Theorem 5.2, we provide a new safety condition, replacing the unbounded SCBF in the original theorem with a bounded SCBF on safety region $\mathcal{C}$.   Then, based on the spatial discretization and Lipschitz continuity, we show in Theorem 5.3 that it is sufficient to check a slightly stricter condition on finite discretized states to establish the safety guarantee on the whole $\mathcal{C}$. We construct neural networks with uniform Lipschitz constant based on the spectral regularization and unconsrtained monotonic neural network (UMNN) to limit the Lipschitz constant in Theorem 5.3. Similarly to the safety guarantee, we further obtain the stability guarantee for NDC and NSC, and we summarize them in Appendix A.1.8.
>
> ---
>
> > **Q2**: It is not comprehensively discussed/explained when to use NSC and when to NDC.
>
> **Response**: Thanks for your comments. We provide a comparison study on NSC and NDC, which is included in Appendix A.3.2. First, the computational complexity of NDC, estimated as $\mathcal{O}((mn)^2)$, is higher than that of NSC, estimated as $\mathcal{O}(mn)$.  Here $m$ represents the batch size and $n$ is the dynamics dimension. Hence, we recommend the application of the NSC in the tasks involving high-dimensional systems. Since the noise brings uncertainty to many systems, compared to NDC, it is more hard to realize the safety guarantee for NSC. As such, the NDC is suitable for dealing with the tasks that need to reduce uncertainties.
>
> ---
>
> Finally, we would like to thank the reviewer again for his/her time and positive feedback on the paper. The revised paper is improved a lot from the theoretical and experimental aspects. We hope that the reviewer will be satisfied with the revised paper and the responses as well, and then consider revising the assessment in support of the revised paper. We may make further improvements according to your feedback.

---

> > ### Author Response · Authors · 2022-11-25
> > **Sincerely Looking Forward to Your Reply**
> >
> > Dear Reviewer cu1i,
> >
> > Thanks again for your time and efforts in reviewing our paper. Your valuable comments and suggestions do help us polish the paper, from the safety and stability guarantees (**Q1**) and the comprehensive discussion on the proposed framework (**Q2**).
> >
> > We are sincerely looking forward to your reply, and we hope you will consider increasing your score if  we have addressed your concerns.
> >
> > Best,
> >
> > Authors

---

> > > ### Comment · Reviewer_cu1i · 2022-12-12
> > > **Response to authors' revision and response**
> > >
> > > Thanks to the authors for their revisions and responses. I think the submission is now better/clearer, and I have raised my score accordingly.

---

> > > > ### Author Response · Authors · 2022-12-12
> > > > **Thank you very much**
> > > >
> > > > We sincerely thank you for increasing the score, and your time and efforts.
> > > >
> > > >
> > > > Best,
> > > >
> > > >
> > > > Authors

---

> > ### Author Response · Authors · 2022-12-02
> > **Friendly reminder: The final stage of the discussion period is drawing to a close**
> >
> > Dear Reviewer cu1i,
> >
> > Thanks again for your comments and suggestions. Your constructive feedback helps us to revise the paper and improve its quality adequately. Since we cannot modify the paper at this stage, we put the latest revision to the anonymous GitHub (see https://anonymous.4open.science/r/SYNC-44D4/1129_full_v1.pdf).
> >
> > This is a friendly reminder that the final stage of the discussion period is drawing to a close. We would appreciate it if you could let us know if your concerns have been addressed and if we can clarify any other concerns.
> >
> > Nevertheless, Thank you for your time and kind efforts.
> >
> > Best,
> >
> > Authors

---

> > ### Author Response · Authors · 2022-12-09
> > **Gentle Reminder: 3 days left in final discussion phase**
> >
> > Dear Reviewer cu1i,
> >
> > Hope you are well! We would like to thank you for your constructive and insightful comments as well as your time and efforts.
> >
> > This is a gentle reminder that there are 3 days remaining in the final discussion phase. We are sincerely looking forward to your reply, and we are more than happy to continue the discussion during the remaining discussion period.
> >
> > Best,
> >
> > Authors

---

### Author Response · Authors · 2022-11-12
**Response to all reviewers**

We would like to thank the reviewers for their time, efforts, and valuable comments and suggestions.  Accordingly, we try our best to make substantial revisions. The common concern is how to rigorously obtain the safety guarantee and stability guarantee for the neural network form functions. According to this instructive concern, we improve the original unrigorous result by providing new theorems to establish the safety and stability guarantee.  Also, the revised article contains the proofs of the new analytical results and the new constructions of neural networks in our framework. Specifically, our main changes are listed as follows.

1. In Theorem 5.2, we provide a new safety condition, replacing the unbounded stochastic control barrier function (SCBF) in the original theorem with a bounded SCBF on safety region $\mathcal{C}$. This condition has low computational complexity in the training stage. The detailed proof for this theorem is included in Appendix A.1.6.
2. In Theorem 5.3, we show that, when the condition on finite training data is satisfied, the safety guarantee on the whole safety region is assured. We provide rigorous proof for this theorem, which is inspired by the existing techniques, such as spatial discretization and Lipschitz continuity [1]. These techniques were used to discuss the discrete-time dynamics; however, we use them to discuss the stochastic delay differential equations (SDDEs). The detailed proof for this theorem is included in Appendix A.1.7.
3. Similar to the safety guarantee, we rigorously establish the results of the stability guarantee for NDC and NSC.
4. Based on the new analytical results, we redefine the loss function with a safety guarantee.
5. We use the spectral regularization [2] to obtain a uniform Lipschitz constant of neural control function. Moreover, we use the unconstrained monotonic neural networks (UMNN) [3] to obtain a class-K function with a uniform Lipschitz constant.
6. We provide more explanations for illustrating Theorem 4.1 and the theorems in Section 5. Also, we depict a diagram for the safety guarantee in Section 5.
7. We reorganize some parts of this article, as suggested by the reviewers, to make the revision more readable. Also, we highlight all the modifications using red colors.

Finally, we thank all the reviewers again for your insightful comments. We do believe that the revised article is much improved not only from the theoretical aspect but also from the experimental aspect.

[1] Lechner, M., Žikelić, Đ., Chatterjee, K., & Henzinger, T. A. (2022, June). Stability verification in stochastic control systems via neural network supermartingales. In Proceedings of the AAAI Conference on Artificial Intelligence (Vol. 36, No. 7, pp. 7326-7336).

[2] Miyato, T., Kataoka, T., Koyama, M., & Yoshida, Y. (2018). Spectral normalization for generative adversarial networks. arXiv preprint arXiv:1802.05957.

[3] Wehenkel, A., & Louppe, G. (2019). Unconstrained monotonic neural networks. Advances in neural information processing systems, 32.

---

### Author Response · Authors · 2022-11-17
**Reminder: The discussion phase is coming to an end**

Dear Reviewers,

Thanks again for your valuable comments. We kindly remind reviewers that the discussion period is coming to an end and we would like to know if there are any remaining changes or questions you would like us to answer. We're more than happy to address these issues (before the rebuttal deadline closes on **November 18th**). Otherwise, could you take this opportunity to re-evaluate our paper and update your score accordingly? Thanks!


Best,

Authors

---

### Decision · Program_Chairs · 2023-01-20

**Decision:**

Accept: poster

**Justification For Why Not Higher Score:**

lack of clarity in writing, especially in expressing certain claims, and limited novelty compared to existing methods

**Justification For Why Not Lower Score:**

The problem studied in the paper is challenging and the proposed solution is interesting and novel.

**Metareview: Summary, Strengths And Weaknesses:**

The reviewers found the problem studied in the paper important and challenging, and the proposed solution interesting and novel. However, there are concerns about 1) certain claims in the paper (e.g., safety guarantee), 2) the lack of clarity in writing, which makes it difficult to properly evaluate the results, 3) novelty compared to existing methods, and 4) some limitations in the experiments. The authors addressed (or partially addressed) some of these concerns during the rebuttal, which eventually convinced the reviewers to increase their score. I would strongly recommend the authors to take the rest of the reviewers' comments into consideration, revise their work, especially better (and more careful) structuring and writing in their next revision.



**Note From Pc:**

if the above contains the word "oral" or "spotlight" please see: "oral" presentation means -> notable-top-5% and "spotlight" means -> notable-top-25%. As stated in our emails, we are disassociating presentation type from AC recommendations